# Distributed State Estimation for Flapping-Wing Micro Air Vehicles with Information Fusion Correction

**DOI:** 10.3390/biomimetics9030167

**Published:** 2024-03-10

**Authors:** Xianglin Zhang, Mingqiang Luo, Simeng Guo, Zhiyang Cui

**Affiliations:** 1School of Aeronautic Science and Engineering, Beihang University, Beijing 100191, China; zxlin727@buaa.edu.cn (X.Z.); cuizhiyang@buaa.edu.cn (Z.C.); 2School of Automation Science and Electrical Engineering, Beihang University, Beijing 100191, China; gsmeng@buaa.edu.cn

**Keywords:** flapping-wing micro air vehicles, distributed state estimation, information fusion correction, non-uniform sampling, Lyapunov–Krasovskii functional

## Abstract

In this paper, we explore a nonlinear interactive network system comprising nodalized flapping-wing micro air vehicles (FMAVs) to address the distributed $H_{\infty }$ state estimation problem associated with FMAVs. We enhance the model by introducing an information fusion function, leading to an information-fusionized estimator model. This model ensures both estimation accuracy and the completeness of FMAV topological information within a unified framework. To facilitate the analysis, each FMAV’s received signal is individually sampled using independent and time-varying samplers. Transforming the received signals into equivalent bounded time-varying delays through the input delay method yields a more manageable and analyzable time-varying nonlinear network error system. Subsequently, we construct a Lyapunov–Krasovskii functional (LKF) and integrate it with the refined Wirtinger and relaxed integral inequalities to derive design conditions for the FMAVs’ distributed $H_{\infty }$ state estimator, minimizing conservatism. Finally, we validate the effectiveness and superiority of the designed estimator through simulations.

## 1. Introduction

Bionics-based FMAV introduces a novel vehicle concept that emulates the flight patterns observed in birds or insects. This type of vehicle not only replicates the agile flight mechanisms found in actual birds and insects, but also demonstrates superior performance in executing intricate and demanding missions [1,2,3,4,5,6]. This enhanced performance is attributed to its high maneuverability, strong adaptability, and low noise characteristics. Furthermore, from a bionic perspective, it is imperative for FMAVs to possess the capability to execute tasks in scenarios characterized by incomplete information.

Most of the current FMAVs primarily concentrate on flapping-wing flight, with limited research dedicated to the distributed state estimation of multiple FMAVs’ post-stabilized flight. Notably, significant advancements have been achieved in the domain of flapping-wing flying vehicles, exemplified by noteworthy creations such as the Nano HummingBird [7] from the United States, SmartBird [8] from Germany, and Dove [9] from the Northwestern Polytechnical University of China. Despite the progress in position control, trajectory tracking, and formation control facilitated by advancements in sensor networks, wireless communication, bionics, and aerodynamics, there remains a substantial gap in exploring the distributed state estimation of multiple FMAVs following stabilized flight [10,11,12,13,14]. In practical applications, the distributed information interaction network among FMAVs is susceptible to information incompleteness phenomena, encompassing missing sampling signals, time delays, and various internal or external interferences. These factors inevitably compromise overall estimation performance, impacting the accuracy and reliability of the entire system [15,16,17,18,19]. Consequently, the effective integration of interaction information within the FMAV network is crucial for achieving efficient, prompt, and reliable distributed state estimation.

State estimation and filtering are integral components of FMAV research, serving as the cornerstone for achieving autonomous flight, trajectory scheduling, tracking, and localization. Therefore, it is imperative to design robust estimators/filters to enhance the system’s stability. Numerous research outcomes related to distributed state estimation and filter design for traditional FMAV information interaction networks, sensor networks, and more, have been documented. Mousavi S.M. et al. [20] introduced an improved adaptive neural network filtering algorithm to rectify the control volume of FMAVs, achieving real-time tracking and state estimation. Liu G et al. [21] devised a multisensor integrated state sensing and estimation method to address the substantial FMAV flight fluctuation problem using Kalman principles, thereby enhancing state estimation accuracy. Yang R et al. [22] proposed a data fusion and attitude estimation algorithm based on the EKF algorithm to counteract instability caused by jitter during FMAV sensor data acquisition’s transient oscillation. He W et al. [23] formulated a state estimator based on uncertain perturbation to address the challenges of unknown time delay and nonlinearity in FMAVs. This design ensures the stability of the bounded control signal and the closed-loop system. Qian W et al. [24] established a multichannel stochastic attack model for various network systems, including FMAVs and sensor networks. They utilized the LKF method to craft a distributed state estimator, satisfying mean-square asymptotic stability for a given $H_{\infty }$ metric. However, many traditional estimator/filter models primarily express the gain of received information of FMAVs by solving a single estimator parameter, resulting in a relatively fixed model. While these models can mitigate the impact of system and external perturbations to some extent—ensuring that the system maintains a normal performance level—they face challenges when the system or its environment becomes harsh. This inevitably leads to performance degradation or destabilization. Consequently, exploring how to leverage known communication information in FMAV networks to maintain superior performance in more general communication environments remains largely uncharted. Building upon this premise, this paper introduces an information fusion correction mechanism within the framework of the classical FMAV distributed state estimator structure. This mechanism enables the fusion of additional state and sampling information during the information interaction process, ensuring the estimation accuracy of the target network.

In the FMAV network system, the acquisition of communication signals, including position and velocity, relies on digital transmission. Before being transmitted to the state estimator, these communication signals received by the FMAV must be obtained as data signals. The traditional approach involves modeling the target flight signal as a discrete-time signal system through a period of uniform sampling. However, this method falls short of capturing the true signal characteristics of FMAV communication, particularly when the sampling period coincides with the signal period [25,26]. This situation proves highly detrimental to the accurate reconstruction of the actual signal. Consequently, the non-uniform sampling method for digital signals is widely adopted. This method not only yields more precise FMAV data signals, but also enhances adaptability when confronting unknown effects. For non-uniform signal sampling, the input delay method proposed in [27] is commonly employed. The main idea is to convert the sampled data system into a continuous-time system with a bounded time delay. For instance, Fridman E et al. [28] and Wang L C et al. [29] proposed data sampling methods for distributed state estimation schemes for time-varying multi-rate systems with channel redundancy and multisensor systems, respectively, and the problem of state estimation for distributed time delay systems based on the input delay method was also addressed in [30]. For the study of non-uniform sampling methods for UAVs, Sun D et al. [31] realizes non-uniform sampling of signals by dividing the aperture of the acquired signals non-uniformly and proposes a method that can ensure that the non-uniform signals have high resolution and consistency. Wang J H et al. [32] introduces time-varying non-uniform communication signals for the topology of UAVs, and utilizes the time delay information and the state information of each intelligent body to establish a discrete formation protocol and gives a sufficient condition for the formation of closed-loop stabilization. Therefore, another objective of this paper is to employ a non-uniform sampling method to ensure a more accurate reconstruction process and enhance the state estimation performance of FMAV communication signals.

Time delay is a prevalent phenomenon in various aircraft control systems, and its existence is a primary contributor to system instability and performance degradation. In the context of FMAV distributed systems, the focus of research has shifted towards stability analysis and estimator design. This shift is necessitated by the impact of the system state influenced by topology, the sampling process, and noise. To ensure asymptotic stabilization of a continuous-time system with bounded time delay under the desired decay index, researchers adopt a dual approach. Firstly, they enhance the information content of the upper and lower bounds of the generalized function by incorporating more time delay information. Secondly, they mitigate conservativeness in the deflation results through the construction of a suitable LKF and the application of a novel integral inequality deflation method. It is crucial to emphasize that achieving less conservative results is not solely dependent on the complexity of the generalized function or the sophistication of the deflation method. Instead, a synergistic collaboration between the two is imperative to enhance result conservatism while considering factors such as computation load and decision variables. In [33], the authors utilized generalized free-weight matrix integral inequalities in conjunction with augmented LKF that feature a pair of integral terms, leading to two stability criteria with superior outcomes. Meanwhile, the authors of [34] employed generalized free-matrix-based integral inequalities alongside the LKF, containing a dual integral, establishing a stability criterion associated with time delay. In another instance, [16] addressed the stability problem in the presence of time delay by constructing an LKF incorporating a time delay dependence matrix. They utilized a single integral inequality based on a relaxation function to derive a new design condition for a distributed H-inf state estimator. Despite the effectiveness of these methods in alleviating conservative conditions, there remains considerable room for improvement, serving as the motivation for our research.

In summary, studying the design of a distributed state estimator based on the non-uniform sampling method and maximizing the utilization of information from FMAV nodes is crucial. In this paper, we aim to establish a distributed state estimator model for information fusion correction and investigate the distributed $H_{\infty }$ state estimation of FMAV network systems under non-uniform sampling. The main highlights of this paper include the following:Improving the traditional distributed state estimator and introducing information fusion correction between nodes in the FMAV network system to enhance estimation accuracy.Equipping each FMAV with an independent sampler, featuring variable sampling periods and utilizing a time delay study method to transform the state estimation problem based on non-uniform sampling into a problem with multiple bounded time delays.Constructing the LKF by fully exploiting time delay information induced by non-uniform sampling. This approach avoids the introduction of complex multiple integrals and generalized functions, effectively reducing the computational burden.Proposing an easy-to-implement distributed $H_{\infty }$ state estimation method with minimal conservatism by employing relaxed and Wirtinger integral inequalities in deflating generalized functions.

The relevant symbol notes used in this chapter are as follows: $R^{n}$ represents n-dimensional Euclidean space. $R^{n\times m}$ is a set of n×m real matrix. $A^{T}$ is the transpose of *A*. P>0 means positive definite matrix. * in the matrix denotes the symmetric element. *I* is an identity matrix with the appropriate dimensions. col{…} is a column vector representing the constituent elements in parentheses. $diag_{N}\left\{\cdot \right\}$ exhibits block diagonal matrix composed of elements in the bracket, $diag_{N}^{i}\left\{A_{i}\right\}=diag_{N}^{i}\left\{A_{1},...,A_{N}\right\}$. A⊗B means the Kronecker product of matrices *A* and *B*.

## 2. Problem Formulation

The system model studied in this paper is the traditional FMAV network system model, which consists of *N* FMAVs. Since the current FMAVs are mainly bionic from the flight mode, i.e., they rely solely on the fluttering wing to generate lift and thrust simultaneously for flight, there is still a big gap between aerodynamic efficiency and birds. In addition, the research on the flight mechanism of birds in terms of sensing, driving, and controlling is still in the primary stage; therefore, if the construction of the system model is convenient, ignoring its complex dynamics and treating all the FMAVs as a single mass point, then the following network system model can be established:(1)$$\begin{array}{c} \left\{\begin{array}{c} \dot{x}\left(t\right)=Ax\left(t\right)+f\left(x\left(t\right)\right)+B\omega \left(t\right)\\ z(t)=Mx(t) \end{array}\right. \end{array}$$
with *N* FMAVs modeled by
(2)$$\begin{array}{c} y_{i}\left(t\right)=C_{i}x\left(t\right)+D_{i}v\left(t\right) \end{array}$$
where $x\left(t\right)\in R^{n}$ is the state vector; $z\left(t\right)\in R^{n}$ is the signal to be estimated; $\omega \left(t\right)\in R^{n}$, $v\left(t\right)\in R^{n}$ is the exogenous disturbance input $\omega \left(t\right),v\left(t\right)\in L_{2}\left[0\;,\;\infty \right)$. $y_{i}\left(t\right)\in R^{n}$ is the measurement output of *i*. System matrices A,B,M, and $C_{i},D_{i}\left(1\leq i\leq N\right)$ are given as constant matrices.

**Assumption** **1**([35])**.**
*The nonlinear function f(x(u)) satisfies $R^{n}\to R^{n},f\left(0\right)=0$, and*
(3)$$\begin{array}{c} {\left[f\left(x\right)-f\left(\varpi \right)-U_{1}\left(x-\varpi \right)\right]}^{T}\left[f\left(x\right)-f\left(\varpi \right)-U_{2}\left(x-\varpi \right)\right]\leq 0\;\;\;\;\forall x,y\in R^{n} \end{array}$$
*where $U_{1},U_{2}\in R^{n\times n}$ are constant matrices describing the linearization direction or error bounds.*

This paper considers a state estimator configuration model with *N* FMAVs, and the topology of the FMAV network system is described by an *N*-order directed graph G=(V,E,A), where $V=\left\{1,2,\ldots ,N\right\}$ is the node set of the FMAVs network, E⊆V×V is the edge set, and $A=[a_{ij}{]}_{N\times N}\left(a_{ij}\geq 0\right)$ is the weighted adjacency matrix. (i,j) indicates an edge in the graph G, and $a_{ij}>0\Leftrightarrow \left(i,j\right)\in U$ indicates that node *i* can accept information from node *j*, and for each i∈V, if there is $a_{ii}=1\;\left(i\in V\right)$, the set of adjacent nodes is recorded as $N_{i}=\left\{j\in V:\left(i,j\right)\in E\right\}$.

In this paper, the communication information around node *i* is shown in Figure 1. According to the given FMAV topology, the information received by node *i* from itself and its neighbor *j* can be expressed as
(4)$$\begin{array}{c} {\bar{y}}_{i}\left(t\right)=\sum_{j\in N_{i}}a_{ij}\left(y_{j}\left(t\right)-C_{j}{\hat{x}}_{j}\left(t\right)\right) \end{array}$$
where ${\bar{y}}_{i}\left(t\right)\in R^{n}$ is the measurement output received by the FMAV *i* from the system.

For each i(1≤i≤N), the sampled signal is generalized by a zero-order hold (ZOH) function where the sequence of hold times is given by $0=t_{0}^{i}<t_{1}^{i}<\ldots <t_{k}^{i}<\ldots$. Then, we have
(5)$$\begin{array}{c} {\hat{y}}_{i}\left(t\right)={\bar{y}}_{i}\left(t_{k}^{i}\right)={\bar{y}}_{i}\left(t-\left(t-t_{k}^{i}\right)\right),\;\;\;\;t_{k}^{i}\leq t<t_{k+1}^{i} \end{array}$$
where ${\hat{y}}_{i}\left(t\right)$ is the true input to the estimator, $t_{k}^{i}$ is the sampling time of the FMAV *i*, and $\lim_{k\to \infty }t_{k}^{i}=\infty$.

Since there is information interaction between each FMAV, considering the information fusion mechanism of each FMAV, it is possible to obtain
(6)$$\begin{array}{c} {\tilde{y}}_{i}\left(t\right)={\hat{y}}_{i}\left(t\right)+H_{i}\varphi_{i}\left(t\right) \end{array}$$
where $\varphi_{i}\left(t\right)$ represents the information fusion from neighbor *j* to *i*, $H_{i}$ represents the gain in information compensation, and
(7)$$\begin{array}{c} \varphi_{i}\left(t\right)=\left[\left(\frac{x\left(t\right)+\sum_{j\in N_{i}}a_{ij}x\left(t-\tau_{i}\left(t\right)\right)}{\sum_{j\in N_{i}}a_{ij}+1}-x\left(t\right)\right)-\left(\frac{{\hat{x}}_{i}\left(t\right)+\sum_{j\in N_{i}}a_{ij}{\hat{x}}_{j}\left(t-\tau_{i}\left(t\right)\right)}{\sum_{j\in N_{i}}a_{ij}+1}-{\hat{x}}_{i}\left(t\right)\right)\right] \end{array}$$

Define $\tau_{i}\left(t\right)\geq t-t_{k}^{i}$ on the interval $t_{k}^{i}\leq t<t_{k+1}^{i}$ and let $\tau_{i}\overset{\Delta }{=}\max\left\{\tau_{i}\left(t\right)\right\}$ be the known scalar of each FMAV i∈V, then the i∈V estimator for each FMAV can be written as
(8)$$\begin{array}{c} \left\{\begin{array}{c} {\dot{\hat{x}}}_{i}\left(t\right)=A{\hat{x}}_{i}\left(t\right)+f\left({\hat{x}}_{i}\left(t\right)\right)+K_{i}{\tilde{y}}_{i}\left(t\right)\\ {\hat{z}}_{i}\left(t\right)=M{\hat{x}}_{i}\left(t\right)\;\;\;\;\;\;\;\;\;t_{k}^{i}\leq t<t_{k+1}^{i} \end{array}\right.\; \end{array}$$
where ${\hat{z}}_{i}\left(t\right)\in R^{n}$ is the estimate for z(t) on the FMAV *i*, and $K_{i}\in R^{n\times n},H_{i}\in R$ is the parameter of state estimator *i* to be determined. It can be easily seen that $\tau_{i}\left(t\right)$ satisfies $\tau_{i}\left(t\right)\in \left[0,\bar{\tau }\right)$ with $\bar{\tau }\overset{\Delta }{=}\max_{i\in V}\left\{\tau_{i}\right\}$ and $t\neq t_{k}^{i}\left(k=0,1,2,...,\infty \right)$. 

**Remark** **1.**
*To ensure that the system maintains excellent estimation performance despite an unstable communication environment, this paper introduces a novel estimator model with an embedded information fusion correction function. This function enhances estimation accuracy by incorporating an information feedback link. Specifically, the traditional signal sampling and estimation link, depicted as the light blue solid line in the upper half of Figure 1, is complemented by the addition of the orange dotted line in the lower right, as illustrated in Figure 1. This addition aims to enhance the utilization rate of multi-node information transmission. The signals sampled by each sampler are consolidated at the data aggregation center and transmitted to the corresponding estimator. Subsequently, all information is fused and calculated by the constructed information fusion correction function $\varphi_{i}\left(t\right)$. The results are then transmitted to each FMAV, where they are used to correct the estimator and refine parameter values. This process enhances the accuracy of the information obtained by the estimator and reduces the conservatism of the estimation error system.*


**Remark** **2.**
*In the existing literature, the prevalent sampling method determines the next sampling moment by providing a time interval, satisfying the formula $\tau_{k}<\bar{\tau }$. However, this approach results in a relatively fixed sampling interval for different communication signals, making it challenging for the signal data to adequately respond to the randomly occurring incomplete information at various nodes in distinct forms and at different times. This limitation hinders the effective reconstruction of real signals. In this paper, we address this issue by adopting a non-uniform sampling method determined by the time-varying function formula $\tau_{i}\left(t\right)$ for different FMAVs. This approach ensures that the sampling intervals of various sampling FMAVs are time-varying and independently bounded.*


Letting the state estimation error be $e_{i}\left(t\right)=x\left(t\right)-{\hat{x}}_{i}\left(t\right)$ and the output estimation error be ${\tilde{z}}_{i}\left(t\right)=z\left(t\right)-{\hat{z}}_{i}\left(t\right)$ for each *i*, we have
(9)$$\begin{array}{c} \left\{\begin{array}{c} \begin{array}{c} {\dot{e}}_{i}\left(t\right)=Ae_{i}\left(t\right)+f\left(e_{i}\left(t\right)\right)+B\omega \left(t\right)-K_{i}\sum_{j\in N_{i}}a_{ij}C_{j}e_{\tau }\\ \;\;\;\;\;\;\;\;\;\;\;\;\;-K_{i}\sum_{j\in N_{i}}a_{ij}D_{j}v_{\tau }-G_{i}\left(\alpha_{i}-1\right)e_{i}\left(t\right)-G_{i}\left(1-\alpha_{i}\right)\sum_{j\in N_{i}}a_{ij}C_{j}e_{\tau } \end{array}\\ {\tilde{z}}_{i}\left(t\right)=Me_{i}\left(t\right) \end{array}\right. \end{array}$$
where $f\left(e_{i}\left(t\right)\right)\overset{\Delta }{=}f\left(x\left(t\right)\right)-f\left({\hat{x}}_{i}\left(t\right)\right),\alpha_{i}=\frac{1}{\sum_{j\in N_{i}}a_{ij}+1},G_{i}=K_{i}H_{i}$.

Setting
$$\begin{array}{ccc} & & \bar{A}\overset{\Delta }{=}{diag}_{N}\left\{A\right\},\;\;\;\bar{B}\overset{\Delta }{=}{diag}_{N}\left\{B\right\},\;\;\;\bar{C}\overset{\Delta }{=}diag_{N}^{i}\left\{C_{i}\right\},\;\;\;\bar{D}\overset{\Delta }{=}diag_{N}^{i}\left\{D_{i}\right\},\\ & & \bar{M}\overset{\Delta }{=}{diag}_{N}\left\{M\right\},\;\;\;\;\bar{K}\overset{\Delta }{=}diag_{N}^{i}\left\{K\right\},\;\;\;\bar{G}\overset{\Delta }{=}diag_{N}^{i}\left\{G\right\},\;\;\;e\left(t\right)\overset{\Delta }{=}col_{N}^{i}\left\{e_{i}\left(t\right)\right\},\\ & & F\left(e\left(t\right)\right)=col_{N}^{i}\left\{f\left(e_{i}\left(t\right)\right)\right\},\;\;\;\tilde{z}\left(t\right)\overset{\Delta }{=}col_{N}^{i}\left\{{\hat{z}}_{i}\left(t\right)\right\},\;\;\;v_{\tau }\left(t\right)\overset{\Delta }{=}col_{N}^{i}\left\{v\left(t-\tau_{i}\left(t\right)\right)\right\},\\ & & e_{\tau }\left(t\right)\overset{\Delta }{=}col_{N}^{i}\left\{e_{i}\left(t-\tau_{i}\left(t\right)\right)\right\},\;\;\;{\bar{\Lambda }}_{\alpha_{i}-1}=diag_{N}^{i}\left\{{\bar{\alpha }}_{i}-1\right\},\;\;\;I_{i}\overset{\Delta }{=}\underset{i-1}{\underset{︸}{0,\ldots ,0}},I,\underset{N-i}{\underset{︸}{0,\ldots ,0}}, \end{array}$$
the estimation error system can be further rewritten into the following form:(10)$$\begin{array}{c} \left\{\begin{array}{c} \begin{array}{c} \dot{e}\left(t\right)=\bar{A}e\left(t\right)+F\left(e\left(t\right)\right)+\bar{B}\omega \left(t\right)-\bar{K}I_{i}\left(A⊗I\right)\bar{C}e_{\tau }\\ \;\;\;\;\;\;\;\;\;\;\;-\bar{K}I_{i}\left(A⊗I\right)\bar{D}v_{\tau }-\bar{G}{\bar{\Lambda }}_{\alpha_{i}-1}e\left(t\right)-\bar{G}{\bar{\Lambda }}_{1-\alpha_{i}}I_{i}\left(A⊗I\right)e_{\tau } \end{array}\\ \bar{z}\left(t\right)=\bar{M}e\left(t\right) \end{array}\right. \end{array}$$

In this paper, we aim to design a set of distributed estimators such that the following two requirements are simultaneously satisfied:The estimated error system is asymptotically stable in the case of v(t)=0 and ω(t)=0;With zero-initial condition, for all nonzero v(t) and ω(t), the output estimation error $\bar{z}\left(t\right)$ satisfies
(11)$$\begin{array}{c} \int_{0}^{\infty }{∥\bar{z}\left(t\right)∥}^{2}dt<\gamma^{2}\int_{0}^{\infty }({∥v(t)∥}^{2}+{∥\omega (t)∥}^{2})dt \end{array}$$
where γ>0 is a prescribed disturbance attenuation level.

Before proceeding, we need the following lemmas in deriving our main results.

**Lemma** **1**([30])**.**
*Under Assumption 1, the following inequalities are obtained:*
$$\begin{array}{c} {\left[\begin{array}{c} e(t)\\ F(e(t)) \end{array}\right]}^{T}\left[\begin{array}{cc} {\tilde{U}}_{1} & {\tilde{U}}_{2}\\ {}* & I \end{array}\right]\left[\begin{array}{c} e(t)\\ F(e(t)) \end{array}\right]\leq 0 \end{array}$$
*where*
$$\begin{array}{c} {\tilde{U}}_{1}\overset{\Delta }{=}\frac{{\hat{U}}_{1}^{T}{\hat{U}}_{2}+{\hat{U}}_{2}^{T}{\hat{U}}_{1}}{2},\;\;\;{\tilde{U}}_{2}\overset{\Delta }{=}-\frac{{\hat{U}}_{1}^{T}+{\hat{U}}_{2}^{T}}{2},\;\;\;{\hat{U}}_{2}\overset{\Delta }{=}diag_{N}\left\{U_{1}\right\},\;\;\;\;{\hat{U}}_{2}\overset{\Delta }{=}diag_{N}\left\{U_{1}\right\}. \end{array}$$

**Lemma** **2**([36])**.**
*For any positive definite matrix $R\in R^{n\times n}$, real scalars a,b satisfying a>b and vector-valued function $x\in \left[a,b\right]\to R^{n}$, the following integral inequalities hold:*
$$\begin{array}{ccc} & & \int_{a}^{b}x^{T}\left(\alpha \right)Rx\left(\alpha \right)d\alpha \geq \frac{1}{b-a}{\left(\int_{a}^{b}x(\alpha )d\alpha \right)}^{T}R\left(\int_{a}^{b}x(\alpha )d\alpha \right)+\frac{3}{b-a}\Omega_{1}^{T}R\Omega_{1}\\ & & \int_{a}^{b}\int_{\beta }^{b}{\dot{x}}^{T}\left(\alpha \right)R\dot{x}\left(\alpha \right)d\alpha \geq 2\Omega_{2}^{T}R\Omega_{2}+4\Omega_{3}^{T}R\Omega_{3} \end{array}$$
*where*
$$\begin{array}{ccc} & & \Omega_{1}=\int_{a}^{b}x(\alpha )d\alpha -\frac{2}{b-a}\int_{a}^{b}\int_{s}^{\beta }x(\alpha )d\alpha d\beta ,\;\;\Omega_{2}=x\left(b\right)-\frac{1}{b-a}\int_{a}^{b}x(\alpha )d\alpha ,\;\;\\ & & \Omega_{3}=x\left(b\right)+\frac{2}{b-a}\int_{a}^{b}x(\alpha )d\alpha -\frac{6}{{\left(b-a\right)}^{2}}\int_{a}^{b}\int_{\beta }^{b}x(\alpha )d\alpha d\beta . \end{array}$$

**Lemma** **3**([37])**.**
*The vector function $x\in \left[0,d_{M}\right]\to R^{n}$, time-varying delay $d\left(t\right)\in \left[0,d_{M}\right]$, symmetric matrix V>0, and any free matrix $T_{1}$ satisfy $\left[\begin{array}{cc} V_{1} & T_{1}\\ {}* & V_{1} \end{array}\right]\geq 0$, where $V_{1}=diag\left\{V,3V\right\}$, we can obtain:*
$$\begin{array}{c} -\int_{t-d(t)}^{t}{\dot{x}}^{T}\left(s\right)V\dot{x}\left(s\right)ds-\int_{t-d_{M}}^{t-d(t)}{\dot{x}}^{T}\left(s\right)V\dot{x}\left(s\right)ds\leq -\frac{1}{d_{M}}\zeta_{1}^{T}\left(t\right){\left[\begin{array}{c} W_{1}\\ W_{2} \end{array}\right]}^{T}\left[\begin{array}{cc} V_{1} & T_{1}\\ {}* & V_{1} \end{array}\right]\left[\begin{array}{c} W_{1}\\ W_{2} \end{array}\right]\zeta_{1}\left(t\right) \end{array}$$
*where*
$$\begin{array}{ccc} & & \zeta_{1}\left(t\right)={\left[x^{T}\left(t\right),x^{T}\left(t-d\left(t\right)\right),x^{T}\left(t-h\right),\upsilon_{1}^{T}\left(t\right),\upsilon_{2}^{T}\left(t\right)\right]}^{T},\\ & & e_{i}=\left[0_{n\times (i-1)n},I,0_{n\times (5-i)n}\right],\;\;i=1,2,...\;,5\\ & & W_{1}=\left[\begin{array}{c} e_{1}-e_{2}\\ e_{1}+e_{2}-2e_{4} \end{array}\right],W_{2}=\left[\begin{array}{c} e_{2}-e_{3}\\ e_{2}+e_{3}-2e_{5} \end{array}\right],\\ & & \upsilon_{1}^{T}\left(t\right)=\frac{1}{d(t)}\int_{t-d(t)}^{t}x(s)ds,\;\upsilon_{2}^{T}\left(t\right)=\frac{1}{h-d(t)}\int_{t-d_{M}}^{t-d(t)}x(s)ds. \end{array}$$

## 3. Main Results

**Theorem** **1.**
*Let the disturbance attenuation level γ>0 be given. For the network system (Equation 1) and FMAVs (Equation 2), the dynamic estimation error system (Equation 10) with double gain (information fusion correction) is asymptotically stable and satisfies the $H_{\infty }$ performance constraint (Equation 11) if there exist scalars $\varepsilon_{1}>0$, $\bar{\tau }>0$ and matrices with $P_{2}>0,\;Q_{i}>0,\;Z_{i}>0,\;R_{i}>0$, diagonal matrix $P_{1}=diag_{N}^{i}\left\{P_{1i}\right\},\;\bar{X}=diag_{N}^{i}\left\{X_{i}\right\},\;\bar{Y}=diag_{N}^{i}\left\{Y_{i}\right\}$, and any free matrices $S_{i}=\left[\begin{array}{cc} S_{1i} & S_{2i}\\ S_{3i} & S_{4i} \end{array}\right]\left(i\in V\right)$ with appropriate dimensions such that the following LMIs hold:*

(12)
$$\begin{array}{ccc} & & \left[\begin{array}{cc} \beta_{i}Z_{\upsilon }+Q_{\upsilon } & S_{i}\\ {}* & Q_{\upsilon } \end{array}\right]>0\;\left(i\in V\right), \end{array}$$


(13)
$$\begin{array}{ccc} & & \left[\begin{array}{cccc} \Theta_{\left[\tau_{i}\left(t\right)=\bar{\tau }\right]} & \Pi_{1} & \Pi_{2} & \Pi_{3}\\ {}* & -\gamma^{2}I & 0 & {\bar{B}}^{T}P_{1}\\ {}* & * & -\gamma^{2}I & -{\bar{X}}_{D}^{T}\\ {}* & * & * & -2P_{1}+L \end{array}\right]<0, \end{array}$$


(14)
$$\begin{array}{ccc} & & \left[\begin{array}{cccc} \Theta_{\left[\tau_{i}\left(t\right)=0\right]} & \Pi_{1} & \Pi_{2} & \Pi_{3}\\ {}* & -\gamma^{2}I & 0 & {\bar{B}}^{T}P_{1}\\ {}* & * & -\gamma^{2}I & -{\bar{X}}_{D}^{T}\\ {}* & * & * & -2P_{1}+L \end{array}\right]<0. \end{array}$$

*where*

$$\begin{array}{ccc} & & \Theta =\left[\begin{array}{cccccccc} \Xi_{11} & \Xi_{12} & \Xi_{13} & \Xi_{14} & \Xi_{15} & \Xi_{16} & \Xi_{17} & 3\bar{R}\\ {}* & \Xi_{22} & \Xi_{23} & 0 & \Xi_{25} & \Xi_{26} & 0 & 24\hat{Z}\\ {}* & * & \Xi_{33} & 0 & \Xi_{35} & 6\bar{Q} & 0 & 0\\ {}* & * & * & -\varepsilon_{1}I & 0 & 0 & 0 & 0\\ {}* & * & * & * & \Xi_{55} & -4{\hat{S}}_{4} & 48\hat{Z} & 0\\ {}* & * & * & * & * & \Xi_{66} & 0 & 48\hat{Z}\\ {}* & * & * & * & * & * & \Xi_{77} & 0\\ {}* & * & * & * & * & * & * & \Xi_{88} \end{array}\right]\\ & & \Xi_{11}=Sym\left\{P_{1}\bar{A}-{\bar{Y}}_{\alpha_{1}}^{T}\right\}+P_{2}-6\sum_{i=1}^{N}Z_{i}-4\sum_{i=1}^{N}\beta_{i}Z_{i}-4\sum_{i=1}^{N}Q_{i}-\varepsilon_{1}{\tilde{W}}_{1}\\ & & \;\;\;\;\;\;\;\;\;\;\;\;+h^{2}\sum_{i=1}^{N}R_{i}+M^{T}M,\\ & & \Xi_{12}=-{\bar{X}}_{C}-{\bar{Y}}_{\alpha_{2}}-2\beta_{i}\bar{Z}-2\bar{Q}-{\bar{S}}_{1}-{\bar{S}}_{2}-{\bar{S}}_{3}-{\bar{S}}_{4},\\ & & \Xi_{13}=\sum_{i=1}^{N}\left(S_{1i}+S_{3i}-S_{2i}-S_{4i}\right),\;\;\Xi_{14}=P_{1}-\varepsilon_{1}{\tilde{W}}_{2},\;\;\Xi_{15}=6\bar{Z}-6\beta_{i}\bar{Z}+6\bar{Q}-4\bar{R},\\ & & \Xi_{16}=2{\bar{S}}_{2}+2{\bar{S}}_{4}-4\bar{R},\;\;\Xi_{17}=3\bar{R}+24\bar{Z},\\ & & \Xi_{22}=-6\hat{Z}-4\beta_{i}\hat{Z}-8\hat{Q}+Sym\left\{{\hat{S}}_{1}+{\hat{S}}_{2}-{\hat{S}}_{3}-{\hat{S}}_{4}\right\},\\ & & \Xi_{23}=-2{\bar{Q}}^{T}-{\tilde{S}}_{1}-{\tilde{S}}_{4}+{\tilde{S}}_{2}+{\tilde{S}}_{3},\;\;\Xi_{25}=6\beta_{i}\hat{Z}+6\hat{Q}+2{\hat{S}}_{3}^{T}+2{\hat{S}}_{4}^{T},\\ & & \Xi_{26}=-2{\hat{S}}_{2}+2{\hat{S}}_{4}+6\hat{Q}-6\hat{Z},\;\;\Xi_{33}=-P_{2}-4\sum_{i=1}^{N}Q_{i},\\ & & \Xi_{35}=-2{\bar{S}}_{3}+2{\bar{S}}_{4},\;\;\Xi_{55}=-18\hat{Z}-12\beta_{i}\hat{Z}-12\hat{Q},\;\;\Xi_{66}=-12\hat{Q}-18\hat{Z},\\ & & \Xi_{77}=-144\hat{Z}-3\hat{R},\;\;\Xi_{88}=-144\hat{Z}-3\hat{R},\\ & & \Pi_{1}={\left[\begin{array}{ccccccccc} B^{T}P & 0 & 0 & 0 & 0 & 0 & 0 & 0 & 0 \end{array}\right]}^{T},\\ & & \Pi_{2}={\left[\begin{array}{ccccccccc} -{\bar{X}}_{D}^{T} & 0 & 0 & 0 & 0 & 0 & 0 & 0 & 0 \end{array}\right]}^{T},\\ & & \Pi_{3}={\left[\begin{array}{ccccccccc} \Pi_{31} & -{\bar{X}}_{C} & 0 & {P_{1}}^{T} & 0 & 0 & 0 & 0 & 0 \end{array}\right]}^{T},\;\;\Pi_{31}=P_{1}A-{\bar{Y}}_{\alpha_{1}},\\ & & \bar{Z}=vec_{N}^{i}\left\{Z_{i}\right\},\;\;\bar{R}=vec_{N}^{i}\left\{R_{i}\right\},\;\;\bar{Q}=vec_{N}^{i}\left\{Q_{i}\right\},\;\;\hat{Z}=diag_{N}^{i}\left\{Z_{i}\right\},\;\;\hat{R}=diag_{N}^{i}\left\{R_{i}\right\},\\ & & \hat{Q}=vec_{N}^{i}\left\{Q_{i}\right\},\;\;{\hat{S}}_{h}=diag_{N}^{i}\left\{S_{hi}\right\},\;\;{\tilde{S}}_{h}=col_{N}^{i}\left\{S_{hi}\right\},\;\;{\bar{S}}_{h}=vec_{N}^{i}\left\{S_{hi}\right\},\left(h=1,\ldots ,4\right),\\ & & Z_{\upsilon }=diag\left\{Z_{i},3Z_{i}\right\},\;\;Q_{\upsilon }=diag\left\{Q_{i},3Q_{i}\right\},\;\;{\bar{X}}_{C}=vec_{N}^{i}\left\{\bar{X}I_{i}\left(A⊗I\right)\bar{C}\right\},\\ & & {\bar{X}}_{D}=vec_{N}^{i}\left\{\bar{X}I_{i}\left(A⊗I\right)\bar{D}\right\},\;\;{\bar{Y}}_{\alpha_{1}}=vec_{N}^{i}\left\{\bar{Y}{\bar{\Lambda }}_{\alpha_{i}-1}I_{i}\right\},\\ & & {\bar{Y}}_{\alpha_{2}}=vec_{N}^{i}\left\{\bar{Y}{\bar{\Lambda }}_{1-\alpha_{i}}I_{i}\left(A⊗I\right)\right\},\;\;L=\sum_{i=1}^{N}{\bar{\tau }}^{2}Q_{i}+\sum_{i=1}^{N}\frac{{\bar{\tau }}^{2}}{2}Z_{i},\;\;\beta_{i}=\frac{\bar{\tau }-\tau_{i}\left(t\right)}{\bar{\tau }}. \end{array}$$


*Moreover, the estimator gain matrices are given by $K_{i}=P_{1i}^{-1}X_{i}$ and $G_{i}=P_{1i}^{-1}Y_{i}.$*


**Proof.** Consider an LKF such that
(15)$$\begin{array}{c} V\left(e\left(t\right)\right)=V_{1}\left(e\left(t\right)\right)+V_{2}\left(e\left(t\right)\right)+V_{3}\left(e\left(t\right)\right) \end{array}$$
where
(16)$$\begin{array}{ccc} & & V_{1}\left(e\left(t\right)\right)=e^{T}\left(t\right)P_{1}e\left(t\right)+\int_{t-\bar{\tau }}^{t}e^{T}\left(s\right)P_{2}e\left(s\right)ds \end{array}$$
(17)$$\begin{array}{ccc} & & V_{2}\left(e\left(t\right)\right)=\sum_{i=1}^{N}\int_{t-\bar{\tau }}^{t}\int_{\theta }^{t}\bar{\tau }{\dot{e}}^{T}\left(s\right)Q_{i}\dot{e}\left(s\right)dsd\theta +\sum_{i=1}^{N}\int_{t-\bar{\tau }}^{t}\int_{\theta }^{t}\bar{\tau }e^{T}\left(s\right)R_{i}e\left(s\right)dsd\theta \end{array}$$
(18)$$\begin{array}{ccc} & & V_{3}\left(e\left(t\right)\right)=\sum_{i=1}^{N}\int_{t-\bar{\tau }}^{t}\int_{\theta }^{t}\int_{\lambda }^{t}{\dot{e}}^{T}\left(s\right)Z_{i}\dot{e}\left(s\right)dsd\lambda d\theta \end{array}$$Calculating the time derivative of V(e(t)) along the trajectory of Equation (Equation 10) with ω(t)=0 and $v_{\tau }\left(t\right)=0$ yields
(19)$$\begin{array}{ccc} & & {\dot{V}}_{1}\left(e\left(t\right)\right)=2e^{T}\left(t\right)P_{1}\dot{e}\left(t\right)+e^{T}\left(t\right)P_{2}e\left(t\right)-e^{T}\left(t-\bar{\tau }\right)P_{2}e\left(t-\bar{\tau }\right)\\ & & {\dot{V}}_{2}\left(e\left(t\right)\right)=\sum_{i=1}^{N}{\bar{\tau }}^{2}{\dot{e}}^{T}\left(t\right)Q_{i}\dot{e}\left(t\right)+\sum_{i=1}^{N}{\bar{\tau }}^{2}e^{T}\left(t\right)R_{i}e\left(t\right) \end{array}$$
(20)$$\begin{array}{ccc} & & \;\;\;\;\;\;\;\;\;\;\;\;\;\;\;\;\;\;\;-\sum_{i=1}^{N}\bar{\tau }\int_{t-\bar{\tau }}^{t}{\dot{e}}^{T}\left(s\right)Q_{i}\dot{e}\left(s\right)ds-\sum_{i=1}^{N}\bar{\tau }\int_{t-\bar{\tau }}^{t}e^{T}\left(s\right)R_{i}e\left(s\right)ds \end{array}$$
(21)$$\begin{array}{ccc} & & {\dot{V}}_{3}\left(e\left(t\right)\right)=\sum_{i=1}^{N}\frac{{\bar{\tau }}^{2}}{2}{\dot{e}}^{T}\left(t\right)Z_{i}\dot{e}\left(t\right)-\sum_{i=1}^{N}\int_{t-\bar{\tau }}^{t}\int_{\theta }^{t}{\dot{e}}^{T}\left(s\right)Z_{i}\dot{e}\left(s\right)dsd\theta \end{array}$$According to the following relationship
(22)$$\begin{array}{ccc} & & \;\;\;\;\int_{t-h}^{t}f(s)ds=\int_{t-h(t)}^{t}f(s)ds+\int_{t-h}^{t-h(t)}f(s)ds\\ & & \;\;\;\;\int_{t-h}^{t}\int_{v}^{t}f(s)dsdv=\int_{t-h(t)}^{t}\int_{v}^{t}f(s)dsdv+\int_{t-h}^{t-h(t)}\int_{v}^{t}f(s)dsdv\\ & & =\int_{t-h(t)}^{t}\int_{v}^{t}f(s)dsdv+\int_{t-h}^{t-h(t)}\int_{t-h(t)}^{t}f(s)dsdv+\int_{t-h}^{t-h(t)}\int_{v}^{t-h(t)}f(s)dsdv \end{array}$$
(23)$$\begin{array}{ccc} & & =\int_{t-h(t)}^{t}\int_{v}^{t}f(s)dsdv+\int_{t-h}^{t-h(t)}\int_{v}^{t-h(t)}f(s)dsdv+\left(h-h\left(t\right)\right)\int_{t-h(t)}^{t}f(s)ds \end{array}$$
we can obtain
$$\begin{array}{ccc} & & -\sum_{i=1}^{N}\int_{t-\bar{\tau }}^{t}\int_{\theta }^{t}{\dot{e}}^{T}\left(s\right)Z_{i}\dot{e}\left(s\right)dsd\theta =-\sum_{i=1}^{N}\int_{t-\tau_{i}\left(t\right)}^{t}\int_{\theta }^{t}{\dot{e}}^{T}\left(s\right)Z_{i}\dot{e}\left(s\right)dsd\theta \end{array}$$
(24)$$\begin{array}{ccc} & & \;\;\;\;\;\;\;\;\;\;\;\;\;\;\;\;\;-\sum_{i=1}^{N}\int_{t-\bar{\tau }}^{t-\tau_{i}\left(t\right)}\int_{\theta }^{t-\tau_{i}\left(t\right)}{\dot{e}}^{T}\left(s\right)Z_{i}\dot{e}\left(s\right)dsd\theta -\left(\bar{\tau }-\tau_{i}\left(t\right)\right)\sum_{i=1}^{N}\int_{t-\tau_{i}\left(t\right)}^{t}{\dot{e}}^{T}\left(s\right)Z_{i}\dot{e}\left(s\right)dsd\theta \;\;\;\;\; \end{array}$$
(25)$$\begin{array}{ccc} & & -\sum_{i=1}^{N}\bar{\tau }\int_{t-\bar{\tau }}^{t}{\dot{e}}^{T}\left(s\right)Q_{i}\dot{e}\left(s\right)ds=-\sum_{i=1}^{N}\bar{\tau }\int_{t-\tau_{i}\left(t\right)}^{t}{\dot{e}}^{T}\left(s\right)Q_{i}\dot{e}\left(s\right)ds-\sum_{i=1}^{N}\bar{\tau }\int_{t-\bar{\tau }}^{t-\tau_{i}\left(t\right)}{\dot{e}}^{T}\left(s\right)Q_{i}\dot{e}\left(s\right)ds \end{array}$$
(26)$$\begin{array}{ccc} & & -\sum_{i=1}^{N}\bar{\tau }\int_{t-\bar{\tau }}^{t}e^{T}\left(s\right)R_{i}e\left(s\right)ds=-\sum_{i=1}^{N}\bar{\tau }\int_{t-\tau_{i}\left(t\right)}^{t}e^{T}\left(s\right)R_{i}e\left(s\right)ds-\sum_{i=1}^{N}\bar{\tau }\int_{t-\bar{\tau }}^{t-\tau_{i}\left(t\right)}e^{T}\left(s\right)R_{i}e\left(s\right)ds \end{array}$$From Lemma 3, the following can be obtained:
(27)$$\begin{array}{ccc} & & \;\;-\left(\bar{\tau }-\tau_{i}\left(t\right)\right)\sum_{i=1}^{N}\int_{t-\tau_{i}\left(t\right)}^{t}{\dot{e}}^{T}\left(s\right)Z_{i}\dot{e}\left(s\right)dsd\theta -\bar{\tau }\sum_{i=1}^{N}\int_{t-\bar{\tau }}^{t}{\dot{e}}^{T}\left(s\right)Q_{i}\dot{e}\left(s\right)dsd\theta \\ & & \leq -\bar{\tau }\sum_{i=1}^{N}\int_{t-\tau_{i}\left(t\right)}^{t}{\dot{e}}^{T}\left(s\right)\left(Q_{i}+\beta_{i}Z_{i}\right)\dot{e}\left(s\right)dsd\theta -\bar{\tau }\sum_{i=1}^{N}\int_{t-\bar{\tau }}^{t-\tau_{i}\left(t\right)}{\dot{e}}^{T}\left(s\right)Q_{i}\dot{e}\left(s\right)dsd\theta \\ & & \leq {\left[\begin{array}{c} W_{1}\\ W_{2} \end{array}\right]}^{T}\left[\begin{array}{cc} -\beta_{i}Z_{\upsilon }-Q_{\upsilon } & -S_{i}\\ {}* & -Q_{\upsilon } \end{array}\right]\left[\begin{array}{c} W_{1}\\ W_{2} \end{array}\right] \end{array}$$
where
$$\begin{array}{c} W_{1}=\left[\begin{array}{c} e\left(t\right)-e(t-\tau_{i}\left(t\right))\\ e\left(t\right)+e\left(t-\tau_{i}\left(t\right)\right)-2\eta_{1}\left(t\right) \end{array}\right],\;\;\;W_{2}=\left[\begin{array}{c} e\left(t-\tau_{i}\left(t\right)\right)-e\left(t-\bar{\tau }\right)\\ e\left(t-\tau_{i}\left(t\right)\right)+e\left(t-\bar{\tau }\right)-2\eta_{2}\left(t\right) \end{array}\right]. \end{array}$$From Lemma 2, it can be determined that
(28)$$\begin{array}{ccc} & & -\sum_{i=1}^{N}\int_{t-\tau_{i}\left(t\right)}^{t}\int_{\theta }^{t}{\dot{e}}^{T}\left(s\right)Z_{i}\dot{e}\left(s\right)dsd\theta \leq -2\sum_{i=1}^{N}\Gamma_{1}^{T}Z_{i}\Gamma_{1}-4\sum_{i=1}^{N}\Gamma_{2}^{T}Z_{i}\Gamma_{2} \end{array}$$
(29)$$\begin{array}{ccc} & & -\sum_{i=1}^{N}\int_{t-\bar{\tau }}^{t-\tau_{i}\left(t\right)}\int_{\theta }^{t-\tau_{i}\left(t\right)}{\dot{e}}^{T}\left(s\right)Z_{i}\dot{e}\left(s\right)dsd\theta \leq -2\sum_{i=1}^{N}\Gamma_{3}^{T}Z_{i}\Gamma_{3}-4\sum_{i=1}^{N}\Gamma_{4}^{T}Z_{i}\Gamma_{4} \end{array}$$
where
$$\begin{array}{ccc} & & \Gamma_{1}=e\left(t\right)-\eta_{1}\left(t\right),\;\;\Gamma_{2}=e\left(t\right)+2\eta_{1}\left(t\right)-6\eta_{3}\left(t\right),\\ & & \Gamma_{3}=e\left(t-\tau_{i}\left(t\right)\right)-\eta_{2}\left(t\right),\;\;\Gamma_{4}=e\left(t-\tau_{i}\left(t\right)\right)+2\eta_{2}\left(t\right)-6\eta_{4}\left(t\right). \end{array}$$From Lemma 2, it can be determined that
(30)$$\begin{array}{ccc} & & -\sum_{i=1}^{N}\bar{\tau }\int_{t-\tau_{i}\left(t\right)}^{t}e^{T}\left(s\right)R_{i}e\left(s\right)ds\leq -\sum_{i=1}^{N}\eta_{1}^{T}\left(t\right)R_{i}\eta_{1}\left(t\right)-3\sum_{i=1}^{N}{\left(\eta_{1}\left(t\right)-\eta_{3}\left(t\right)\right)}^{T}R_{i}\left(\eta_{1}\left(t\right)-\eta_{3}\left(t\right)\right)\;\;\;\;\;\;\; \end{array}$$
(31)$$\begin{array}{ccc} & & -\sum_{i=1}^{N}\bar{\tau }\int_{t-\bar{\tau }}^{t-\tau_{i}\left(t\right)}e^{T}\left(s\right)R_{i}e\left(s\right)ds\leq -\sum_{i=1}^{N}\eta_{2}^{T}\left(t\right)R_{i}\eta_{2}\left(t\right)-3\sum_{i=1}^{N}{\left(\eta_{2}\left(t\right)-\eta_{4}\left(t\right)\right)}^{T}R_{i}\left(\eta_{2}\left(t\right)-\eta_{4}\left(t\right)\right)\;\;\;\;\;\;\; \end{array}$$
where
$$\begin{array}{ccc} & & \eta_{1}\left(t\right)=\frac{1}{\tau_{i}\left(t\right)}\int_{t-\tau_{i}\left(t\right)}^{t}e(s)ds,\;\;\eta_{2}\left(t\right)=\frac{1}{t-\tau_{i}\left(t\right)}\int_{t-\bar{\tau }}^{t-\tau_{i}\left(t\right)}e(s)ds,\\ & & \eta_{3}\left(t\right)=\frac{1}{{\left(\tau_{i}\left(t\right)\right)}^{2}}\int_{t-\tau_{i}\left(t\right)}^{t}\int_{\theta }^{t}e(s)ds,\;\;\eta_{4}\left(t\right)=\frac{1}{{\left(t-\tau_{i}\left(t\right)\right)}^{2}}\int_{t-\bar{\tau }}^{t-\tau_{i}\left(t\right)}\int_{\theta }^{t-\tau_{i}\left(t\right)}e(s)ds \end{array}$$**Remark** **3.**
*Given that the estimator structure employed in this paper incorporates FMAV network topology information, and the sampling interval of each vehicle varies over time, the construction of the generalized function and the deflation method of the integration term play a crucial role in determining the complexity and conservatism of the results. To enhance computational efficiency, we introduce single and double integration terms in the construction of the generalized function, making full use of the time delay information introduced by non-uniform sampling. This approach avoids the introduction of complex multiple integration and generalization terms, effectively sidestepping unnecessary computational burden. Specifically, when dealing with the integral terms $\sum_{i=1}^{N}\bar{\tau }\int_{t-\bar{\tau }}^{t}e^{T}\left(s\right)Q_{i}e\left(s\right)ds$ and $\sum_{i=1}^{N}\bar{\tau }\int_{t-\bar{\tau }}^{t}e^{T}\left(s\right)R_{i}e\left(s\right)ds$ in ${\dot{V}}_{2}\left(e\left(t\right)\right)$, we employ time delay segmentation to obtain $\sum_{i=1}^{N}\bar{\tau }\int_{t-\tau i(t)}^{t}e^{T}\left(s\right)Q_{i}e\left(s\right)ds$, $\sum_{i=1}^{N}\bar{\tau }\int_{t-\bar{\tau }}^{t-\tau_{i}\left(t\right)}e^{T}\left(s\right)Q_{i}e\left(s\right)ds$, $\sum_{i=1}^{N}\bar{\tau }\int_{t-\tau i(t)}^{t}e^{T}\left(s\right)R_{i}e\left(s\right)ds$, and $\sum_{i=1}^{N}\bar{\tau }\int_{t-\bar{\tau }}^{t-\tau_{i}\left(t\right)}e^{T}\left(s\right)R_{i}e\left(s\right)ds$. Subsequently, estimation is carried out by Lemma 2, producing four vectors, $\eta_{1}\left(t\right)$, $\eta_{2}\left(t\right)$, $\eta_{3}\left(t\right)$, and $\eta_{4}\left(t\right)$, which are influenced by the number of FMAVs in the network and change over time. The intersection relationship between the four vectors and the state vectors e(t), $e(t-\tau_{i}\left(t\right))$, and $e(t-\bar{\tau })$ is established through the introduction of the free matrix $S_{i}$. Compared with the Jensen inequality, commonly used in existing distributed state estimation, the deflation accuracy of the generalized derivatives is improved, contributing to reduced conservatism. Furthermore, compared with free-matrix- and auxiliary-function-based methods utilized in the stability analysis of time delay systems, the introduction of decision variables is significantly reduced, resulting in reduced complexity.*
Considering Lemma 1 and Equations (Equation 10), (Equation 19)–(Equation 31), we have
(32)$$\begin{array}{ccc} \dot{V}\left(e\left(t\right)\right) & & \leq \varpi^{T}\left(t\right)\Theta_{1}\varpi \left(t\right)-\varepsilon_{1}{\left[\begin{array}{c} e(t)\\ F(e(t)) \end{array}\right]}^{T}\left[\begin{array}{cc} {\tilde{U}}_{1} & {\tilde{U}}_{2}\\ {}* & I \end{array}\right]\left[\begin{array}{c} e(t)\\ F(e(t)) \end{array}\right]+{\dot{e}}^{T}\left(t\right)L\dot{e}\left(t\right)\\ & & \leq \varpi^{T}\left(t\right)\tilde{\Theta }\varpi \left(t\right)+{\dot{e}}^{T}\left(t\right)L\dot{e}\left(t\right) \end{array}$$
where
$$\begin{array}{ccc} & & \tilde{\Theta }=\left[\begin{array}{cccccccc} {\tilde{\Xi }}_{11} & \Xi_{12} & \Xi_{13} & \Xi_{14} & \Xi_{15} & \Xi_{16} & \Xi_{17} & 3\bar{R}\\ {}* & \Xi_{22} & \Xi_{23} & 0 & \Xi_{25} & \Xi_{26} & 0 & 24\hat{Z}\\ {}* & * & \Xi_{33} & 0 & \Xi_{35} & 6\bar{Q} & 0 & 0\\ {}* & * & * & -\varepsilon_{1}I & 0 & 0 & 0 & 0\\ {}* & * & * & * & \Xi_{55} & -4{\hat{S}}_{4} & 48\hat{Z} & 0\\ {}* & * & * & * & * & \Xi_{66} & 0 & 48\hat{Z}\\ {}* & * & * & * & * & * & \Xi_{77} & 0\\ {}* & * & * & * & * & * & * & \Xi_{88} \end{array}\right]\\ & & {\tilde{\Xi }}_{11}=Sym\left\{P_{1}\bar{A}-{\bar{G}}_{\alpha_{1}}^{T}P_{1}\right\}+P_{2}-6\sum_{i=1}^{N}Z_{i}-4\beta \sum_{i=1}^{N}Z_{i}-4\sum_{i=1}^{N}Q_{i}-\varepsilon_{1}{\tilde{W}}_{1}+h^{2}\sum_{i=1}^{N}R_{i},\\ & & \varpi \left(t\right)=\left[\begin{array}{cccccccc} e^{T}\left(t\right) & e_{\tau }^{T}\left(t\right) & e^{T}\left(t-\bar{\tau }\right) & F(e(t)) & \eta_{1}\left(t\right) & \eta_{2}\left(t\right) & \eta_{3}\left(t\right) & \eta_{4}\left(t\right) \end{array}\right] \end{array}$$By using the Schur complement, it follows from Equations (13) and (14) that $\dot{V}\left(e\left(t\right)\right)<0$, which means that the estimation error Equation (Equation 10) with ω(t)=0 and $v_{\tau }\left(t\right)=0$ is asymptotically stable.Let us now move to the $H_{\infty }$ performance analysis for the estimation error Equation (Equation 10). For all nonzero $\omega \left(t\right)\in L_{2}\left[0,\infty \right)$ and $v_{\tau }\left(t\right)\in L_{2}\left[0,\infty \right)$, it can be determined from Equations (Equation 10) and (Equation 32) that
(33)$$\begin{array}{c} V\left(e\left(t\right)\right)+{∥\tilde{z}\left(t\right)∥}^{2}-\gamma^{2}{∥\omega (t)∥}^{2}-\gamma {∥v_{\tau }\left(t\right)∥}^{2}\leq \varsigma^{T}\left(t\right)\hat{\Phi }\varsigma \left(t\right)+{\dot{e}}^{T}\left(t\right)L\dot{e}\left(t\right) \end{array}$$
where
$$\begin{array}{ccc} & & \varsigma \left(t\right)\overset{\Delta }{=}\left[\begin{array}{ccc} \varpi^{T}\left(t\right) & \omega^{T}\left(t\right) & v_{\tau }^{T}\left(t\right) \end{array}\right],\;\;\hat{\Phi }=\left[\begin{array}{ccc} \Phi & \Pi_{1} & \Pi_{2}\\ {}* & -\gamma^{2}I & 0\\ {}* & * & -\gamma^{2}I \end{array}\right]. \end{array}$$By using the Schur complement, we can obtain
(34)$$\begin{array}{c} \begin{array}{c} \left[\begin{array}{cccc} \Theta & \Pi_{1} & \Pi_{2} & \Pi_{3}\\ {}* & -\gamma^{2}I & 0 & {\bar{B}}^{T}P_{1}\\ {}* & * & -\gamma^{2}I & -{\bar{X}}_{D}^{T}\\ {}* & * & * & -P_{1}L^{-1}P_{1} \end{array}\right]<0, \end{array} \end{array}$$Furthermore, it can be derived from Equations (13) and (14) that
(35)$$\begin{array}{c} \left(V\left(e\left(t\right)\right)+{∥\tilde{z}\left(t\right)∥}^{2}-\gamma^{2}{∥\omega (t)∥}^{2}-\gamma {∥v_{\tau }\left(t\right)∥}^{2}\right)<0 \end{array}$$
for all nonzero ω(t) and $v_{\tau }\left(t\right)$. Consequently, we obtain
(36)$$\begin{array}{ccc} & & J\left(t\right)=\int_{0}^{t}\left({∥\tilde{z}\left(s\right)∥}^{2}-\gamma^{2}{∥\omega (s)∥}^{2}-\gamma^{2}{∥v_{\tau }\left(s\right)∥}^{2}+\dot{V}\left(e\left(s\right)\right)\right)ds-V\left(e\left(t\right)\right)+V\left(e\left(0\right)\right)\\ & & \;\;\;\;\;\;\leq \int_{0}^{t}\left({∥\tilde{z}\left(s\right)∥}^{2}-\gamma^{2}{∥\omega (s)∥}^{2}-\gamma^{2}{∥v_{\tau }\left(s\right)∥}^{2}+\dot{V}\left(e\left(s\right)\right)\right)ds<0. \end{array}$$By considering the zero-initial value, it is easily known that J(t)<0, then the $H_{\infty }$ performance constraint Equation (Equation 11) is immediately satisfied. Meanwhile, by using the inequality $-P_{1}L^{-1}P_{1}<-2P_{1}+L$, the LMIs Equations (13) and (14) in Theorem 1 are guaranteed by LMI (Equation 34). The proof of this theorem is now complete. □

When only single gain is considered, we can obtain the following estimated error system
(37)$$\begin{array}{c} \left\{\begin{array}{c} \dot{e}\left(t\right)=\bar{A}e\left(t\right)+F\left(e\left(t\right)\right)+\bar{B}\omega \left(t\right)-\bar{K}I_{i}\left(A⊗I\right)\bar{C}e_{\tau }-\bar{K}I_{i}\left(A⊗I\right)\bar{D}v_{\tau }\\ \bar{z}\left(t\right)=\bar{M}e\left(t\right) \end{array}\right. \end{array}$$
and derive the following corollary.

**Corollary** **1.**
*Let the disturbance attenuation level γ>0 be given. For the network system (Equation 1) and FMAVs Equation (Equation 2), the dynamic estimation error system Equation (Equation 37) with single gain (traditional method) is asymptotically stable and satisfies the $H_{\infty }$ performance constraint Equation (Equation 11) if there exist scalars $\varepsilon_{1}>0$, $\bar{\tau }>0$ and symmetric matrix $P_{2}>0,\;Q_{i}>0,\;Z_{i}>0,\;R_{i}>0$, diagonal matrix $P_{1}=diag_{N}^{i}\left\{P_{1i}\right\},\;\bar{X}=diag_{N}^{i}\left\{X_{i}\right\}$, and any free matrices $S_{i}=\left[\begin{array}{cc} S_{1i} & S_{2i}\\ S_{3i} & S_{4i} \end{array}\right]\left(i\in V\right)$ of appropriate dimensions such that Equations (Equation 12)–(Equation 14) with $Y_{i}=0$ are feasible.*

*Moreover, the estimator gain matrix is given by ${\hat{K}}_{i}=P_{1i}^{-1}X_{i}$.*


## 4. Simulations

In this paper, five FMAVs are simulated to analyze the system’s state. Each FMAV employs distributed communication within the system, and the communication topology between them is depicted in Figure 2. To streamline the simulation calculation, the simulation focuses solely on the XOY transverse plane of mass movement. The system matrix of the model is as follows:$$\begin{array}{c} A=\left[\begin{array}{cc} -0.6 & 0.2\\ 0 & -0.8 \end{array}\right],B={\left[\begin{array}{cc} 0.5 & 1 \end{array}\right]}^{T},M=\left[\begin{array}{cc} 0.1 & 0.1 \end{array}\right]. \end{array}$$

Let the nonlinear functions f(x(t)) be selected as
$$\begin{array}{c} f\left(x\left(t\right)\right)=0.5\left(\left(U_{1}+U_{2}\right)x\left(t\right)+\left(U_{2}-U_{1}\right)\sin\left(t\right)x\left(t\right)\right) \end{array}$$
where
$$\begin{array}{c} U_{1}=\left[\begin{array}{cc} 0.2 & 0\\ 0 & -0.15 \end{array}\right],\;\;U_{2}=\left[\begin{array}{cc} 0.15 & 0\\ 0 & -0.18 \end{array}\right] \end{array}$$

Consider the following FMAV network:$$\begin{array}{ccc} & & C_{1}=\left[\begin{array}{cc} 0.1 & 0 \end{array}\right],C_{2}=\left[\begin{array}{cc} 0.2 & 0.1 \end{array}\right],C_{3}=\left[\begin{array}{cc} 0.5 & 0.7 \end{array}\right],C_{4}=\left[\begin{array}{cc} -0.1 & 0.2 \end{array}\right],\\ & & C_{5}=\left[\begin{array}{cc} -0.2 & 0.1 \end{array}\right],D_{1}=1,D_{2}=0.5,D_{3}=0.7,D_{4}=-0.5,D_{5}=-0.1. \end{array}$$

The topology of FMAV network is represented by a graph G=(V,E,A) with the set of nodes V={1,2,3,4,5}, the set of edges of E={(1,1),(1,2),(1.4),(2,2),(2,4),(2,5),(3,1),(3,2),(3,3),(4,3),(4,4),(4,5),(5,1),(5,3),(5,5)}, and the following adjacency matrix $A=\left[\begin{array}{ccccc} 1 & 1 & 0 & 1 & 0\\ 0 & 1 & 0 & 1 & 1\\ 1 & 1 & 1 & 0 & 0\\ 0 & 0 & 1 & 1 & 1\\ 1 & 0 & 1 & 0 & 1 \end{array}\right]$.

The upper bounds of the sampling time for each FMAV are taken as $\tau_{1}=0.15,\;\tau_{2}=0.18$, $\tau_{3}=0.17,\;\tau_{4}=0.20,\;\tau_{5}=0.18$, where the non-uniform sampling interval is shown in Figure 3. Respectively, we then have $\bar{\tau }=0.20$. According to Theorem 1, the following distributed state estimation parameters can be obtained:
$$\begin{array}{ccc} & & K_{1}=\left[\begin{array}{c} 0.0613\\ -0.0086 \end{array}\right],\;K_{2}=\left[\begin{array}{c} 0.1637\\ 0.0633 \end{array}\right],\;K_{3}=\left[\begin{array}{c} 0.0147\\ -0.0203 \end{array}\right],\\ & & K_{4}=\left[\begin{array}{c} 0.4115\\ -0.2534 \end{array}\right],\;K_{5}=\left[\begin{array}{c} 0.3439\\ -0.1601 \end{array}\right],\\ & & G_{1}=\left[\begin{array}{c} -2.0553\\ -1.1338 \end{array}\right],\;G_{2}=\left[\begin{array}{c} -1.5527\\ -0.6826 \end{array}\right],\;G_{3}=\left[\begin{array}{c} -2.2523\\ -1.3756 \end{array}\right],\\ & & G_{4}=\left[\begin{array}{c} -2.0324\\ -1.2511 \end{array}\right],\;G_{5}=\left[\begin{array}{c} -1.8740\\ -1.1503 \end{array}\right]. \end{array}$$

And according to Corollary 1, we can obtain
$$\begin{array}{ccc} & & {\hat{K}}_{1}=\left[\begin{array}{c} -0.1087\\ -0.0698 \end{array}\right],\;{\hat{K}}_{2}=\left[\begin{array}{c} 0.0231\\ -0.0045 \end{array}\right],\;{\hat{K}}_{3}=\left[\begin{array}{c} -0.0728\\ -0.0703 \end{array}\right],\\ & & {\hat{K}}_{4}=\left[\begin{array}{c} 0.0057\\ -0.0472 \end{array}\right],\;{\hat{K}}_{5}=\left[\begin{array}{c} 0.0148\\ -0.02 \end{array}\right]. \end{array}$$

Additionally, the optimal performance levels specified in Equation (Equation 11) are $\gamma_{T_{1}}^{*}=2.017$ and $\gamma_{C_{1}}^{*}=2.036$, respectively, which shows that the system is more resistant to perturbation under information fusion correction. In the simulation, to better approximate real-world scenarios, the system noise is chosen as a periodic noise that diminishes over time, represented by $\omega \left(t\right)=e^{-0.2t}\sin\left(t\right)$. Meanwhile, the measurement noise is selected to mimic the potential high-frequency vibrations encountered during signal transmission, denoted as $v\left(t\right)=\frac{\sin(10t+1)}{3t+1}$. The initial conditions are set as $x\left(0\right)={\left[\begin{array}{cc} 0.3 & -0.2 \end{array}\right]}^{T}$, and ${\hat{x}}_{i}\left(0\right)\left(i=1,2,3,4,5\right)$ are chosen randomly according to the monomorphic distribution on ${\left[\begin{array}{cc} 0.3 & -0.2 \end{array}\right]}^{T}$.

The simulation results are depicted in Figure 4, Figure 5, Figure 6 and Figure 7. Among them, Figure 4 and Figure 5 display the estimation errors under the traditional estimator model and the information fusion correction model, respectively. It is evident that the peak phase errors of each FMAV are reduced, with a maximum shrinkage of 16.57% and an average decrease of 7.3% (see Table 1). Figure 6 and Figure 7 showcase the estimation outputs of the two estimation methods, respectively. The mean value of the output signals under the corrective estimation converges to ±0.001 compared with the traditional method within 0.6s. Furthermore, both figures reveal that the estimation errors of FMAVs and the output signals of the system are more centralized. This observation means that the FMAV’s estimation of the system signals is faster, exhibiting better consistency and synergy. When multiple FMAVs are performing cooperative missions and flying in formation, the method proposed in this paper can be effectively utilized to cope with the time delay that occurs randomly during the communication process and to improve the topological information utilization by introducing the information fusion correction function. Thus, it can alleviate the perturbing effects of the current non-constant aerodynamic mechanism of the flapping-wing vehicle and the strong coupling between the aerodynamics and structure of the flapping wing on the state estimation, and improve the estimation accuracy of the system as a whole.

## 5. Conclusions

In this paper, we investigate the distributed $H_{\infty }$ state estimation problem for FMAVs under non-uniform sampling. We enhance the traditional distributed state estimator model by introducing an information fusion function to correct the communication information in the FMAVs’ network system. This improvement aims to enhance the accuracy of signal transmission while ensuring estimation performance. Utilizing the Lyapunov stability principle and the method of linear matrix inequality, we provide criteria and a parameter design method for the distributed state estimator to satisfy the $H_{\infty }$ performance index. Our approach combines the Lyapunov–Krasovskii functional (LKF) and deflation methods, along with cooperation with the information fusion function correction. This combination significantly reduces conservatism in the results and enhances the convergence speed of the outcomes. Finally, the designed estimator’s effectiveness and superiority are validated through simulation. In our upcoming study, we aim to enhance the system model by thoroughly analyzing the aerodynamics of FMAVs. We plan to incorporate information fusion correction into the framework to address various randomly occurring information incompleteness phenomena. Additionally, we will explore novel approaches to minimize the conservatism of conclusions in the distributed state estimation of FMAVs.

## Figures and Tables

**Figure 1 biomimetics-09-00167-f001:**
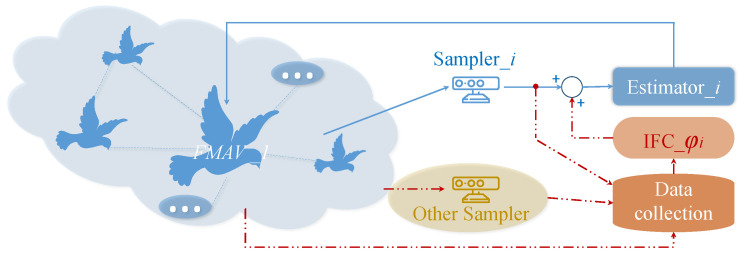
Local structure of FMAV *i* information fusion correction.

**Figure 2 biomimetics-09-00167-f002:**
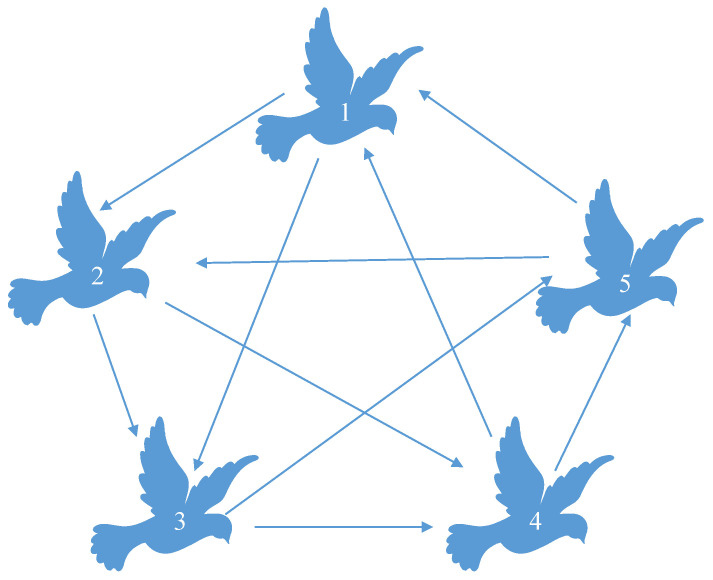
Topological structure of the FMAVs.

**Figure 3 biomimetics-09-00167-f003:**
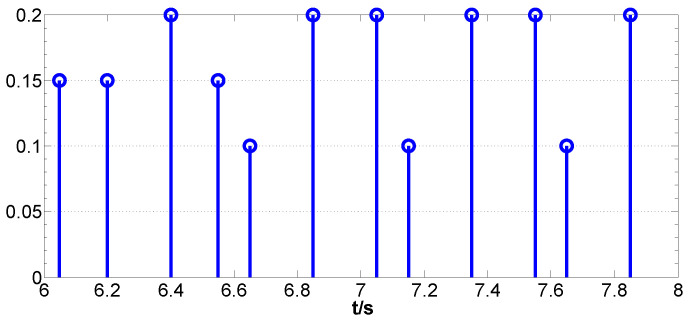
Non-uniform sampling intervals (local).

**Figure 4 biomimetics-09-00167-f004:**
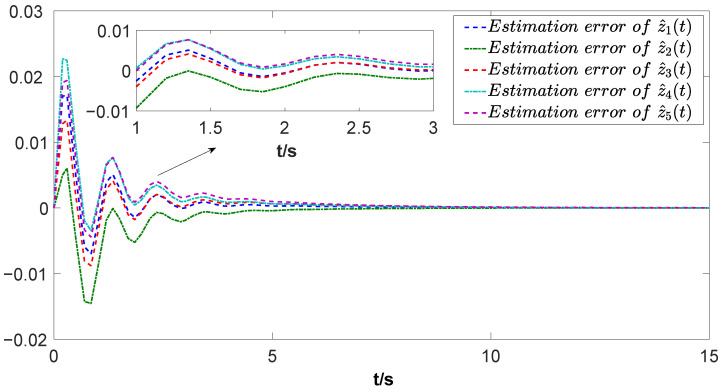
Estimation error under the traditional method.

**Figure 5 biomimetics-09-00167-f005:**
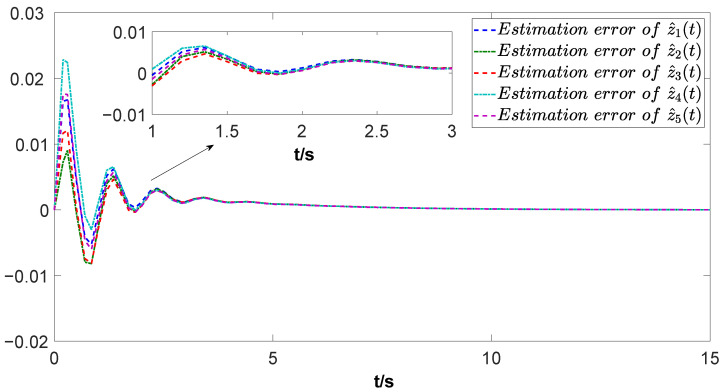
Estimation error under the information fusion correction.

**Figure 6 biomimetics-09-00167-f006:**
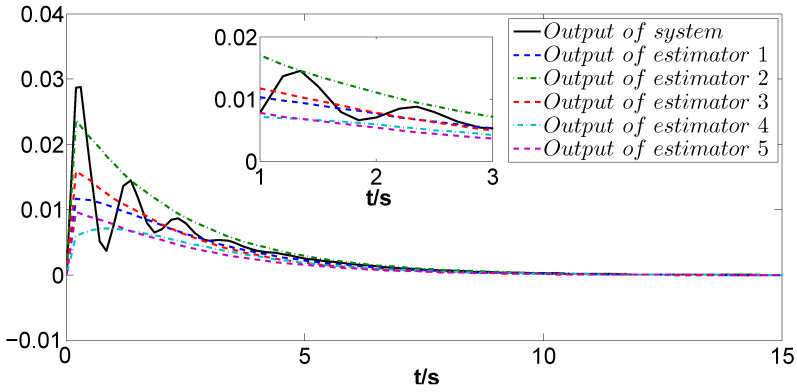
Outputs of z(t) and $z_{i}\left(t\right)$ under the traditional method.

**Figure 7 biomimetics-09-00167-f007:**
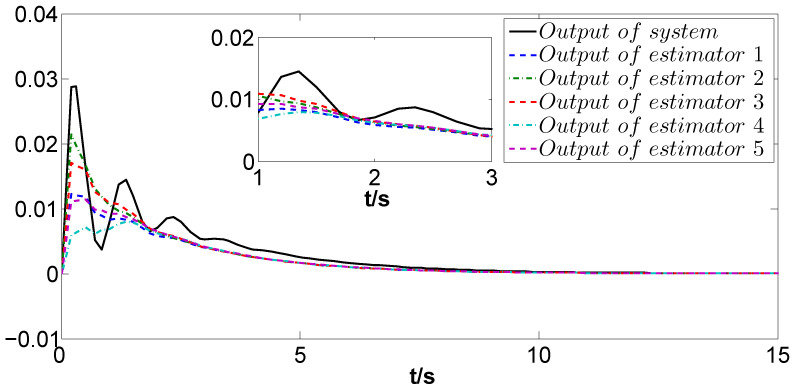
Outputs of z(t) and $z_{i}\left(t\right)$ under the information fusion correction.

**Table 1 biomimetics-09-00167-t001:** Peak phase and error decrease ratio for each FMAV.

	${\mathrm{FMAV}}_{1}$	${\mathrm{FMAV}}_{2}$	${\mathrm{FMAV}}_{3}$	${\mathrm{FMAV}}_{4}$	${\mathrm{FMAV}}_{5}$
$H_{C}$	0.017	0.006	0.013	0.023	0.019
$L_{C}$	−0.007	−0.015	−0.009	−0.003	−0.004
$H_{T}$	0.017	0.009	0.012	0.023	0.018
$L_{T}$	−0.005	−0.008	−0.008	−0.003	−0.006

## Data Availability

The data used to support the findings of this study are available from the corresponding author upon request.

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
