# Peer review of "Distributed State Estimation for Flapping-Wing Micro Air Vehicles with Information Fusion Correction"

_biomimetics, 2024, doi:10.3390/biomimetics9030167_

Round 1
Reviewer 1 Report
Comments and Suggestions for Authors
1- The mathematical process process is quite detailed and as far as I could follow them looks fine, nonetheless, for such a detailed mathematical process, not enough simulations have been provided. Such limited cases of simulation raises great concern regarding the quality of the work. I suggest authors to shorten the mathematical process and add more case-studies.
2- It so appears that the dynamics of the FMAVs has no role in the presented mathematical process. That is, there is no restriction or concern regarding the dynamic characteristics of the network of birds; specifically A, B and M in the system model. It is hard to believe such an indirect claim, authors need to clarify this issue. Otherwise, one may conclude that the presented process is applicable to a fighters formation flight !
3- I am also concerned with the nature of the control problem of a network of flying MAVs in the real-world. That is, authors need to demonstrate the significance of their work in a more realistic control problem. Graphs of 4 to 7 hardly demonstrate the suitability of the method.
4- I am also concerned about two parameters mentioned in the paper, that is "Sampling " and "Delay" I suggest Authors discuss the role of these two parameters
Reviewer 2 Report
Comments and Suggestions for Authors
I didn't validate the mathematical derivations, however, there are fundamental questions that need to be addressed before publishing this paper. Following are my comments:
How is this study related to flapping wing micro air vehicles? This is more like a topology of any vehicle.
No discussion was found describing a topology of flapping wing micro air vehicles and the challenges associated with it.
The problem formulation needs to be rewritten for the flapping wing scenario and needs to highlight the elements that are more specific to flapping wing micro air vehicles.
What are those states that need to be corrected for flapping wings?
What are the sensors required to get those states and how does this methodology improve overall performance for flapping wing microvehicle applications?
What are the practical limitations associated with state estimation of flapping wings and how the proposed study improve it?
I recommend a systematic discussion on flapping wing micro air vehicles and their topology. What specific issues arise in a topology of flapping wing micro air vehicles in comparison to a topology of other air vehicles?
In the result section it is required to discuss how flapping wing micro air vehicles related specific issues are resolved with the proposed methodology.
Comments on the Quality of English LanguageThe quality of the language is appropriate.
Reviewer 3 Report
Comments and Suggestions for Authors
This paper presents an information fusion based estimator model for distributed state estimation of Flapping-wing Micro Air Vehicles (FMAVs). The designed estimator is demonstrated through simulations.
The variables x and omega are of different dimensions according to the definitions below Eq(2). Hence, Eq(3) is incorrect. The authors need to check this.
How are the function f and the constant matrices U1 and U2 determined? Some discussions should be given.
In the simulation example, measurement noise should be considered.
Reviewer 4 Report
Comments and Suggestions for Authors
In this paper, the authors study a nonlinear interactive networked system consisting of node-based flapping wing micro air vehicles (FMAVs) to solve a distributed H∞ state estimation problem associated with FMAVs. The authors extend the model by introducing an information fusion function, which leads to an information fusion-based estimator model. This model ensures both the accuracy of the estimator and the completeness of the topological information of the FMAV in a unified framework. To facilitate analysis, each received FMAV signal is individually sampled using independent and time-varying samplers. Converting the received signals into equivalent bounded time-varying delays using the input delay method yields a more manageable and analyzable time-varying nonlinear network error system. The authors then construct the Lyapunov-Krasovskii functional (LKF) and integrate it with the refined Wirtinger and Relax integral inequalities to obtain design conditions for a distributed H∞ state estimator FMAV that minimizes conservatism. Finally, the authors validate the effectiveness and superiority of the designed estimator through simulations.
My comments:
1) English is very bad. For example Line 120 "...are know constant matrices..."
2) Lines 106-107: The sign "<" is said to imply negative definiteness for a matrix, but on Line 235 this is applied to the derivative of the Lyapunov function, which is obviously a scalar
3) Simulation section: Generally, it is not clear why this particular FMAV model is used, where it came from (Lines 264-267)
Round 2
Reviewer 3 Report
Comments and Suggestions for Authors
The authors have adequately addressed my comment.
Reviewer 4 Report
Comments and Suggestions for Authors
The authors have revised according to comments